# The flow experience: Polish adaptation and validation of the psychological flow scale (PFS)

Marcin Wojtasiński[1]*, Przemysław Tużnik[1], Tomasz Jankowski[1], Silvia Leoni[2], Mateusz Chwaszcz[1], Dorota Miszczyszyn[1], Maria Banasik[1], Paweł Augustynowicz[1]

1 Institute of Psychology, John Paul II Catholic University of Lublin (KUL), Lublin, Poland, 2 Department of Economics and Management, University of Florence, Florence, Italy

* marcin.wojtasinski@kul.pl

## Abstract

Flow is an absorbing, effortless, and intrinsically rewarding state that unfolds over time. We adapted the nine-item Psychological Flow Scale (PFS) to Polish and evaluated it in a preregistered laboratory study designed to capture fine-grained changes in flow. After individual skill calibration, participants completed a 20-trial pursuit-tracking task following a chaotic Lorenz trajectory; data from 140 participants met inclusion criteria. Multilevel confirmatory factor analysis supported the theorized structure: Absorption, Effortless Control, and Intrinsic Reward formed correlated first-order factors nested under a second-order Flow factor at both the within-person and between-person levels ($\chi^2(46) = 345.35$, CFI = 0.98, RMSEA = 0.05). Reliability was excellent for aggregated scores (generalizability RkF = 1.00) and remained high for detecting trial-to-trial change (Rc = 0.88), indicating sensitivity to momentary fluctuations. Convergent validity was evidenced by moderate correlations with the Flow Short Scale administered concurrently (r = .28–.39), low-to-modest correlations with task performance score (r = .13–.32), and low-to-modest associations with the General Flow Proneness Scale (r = .13–.26). Complementary hierarchical exploratory graph analysis corroborated this three-facet-plus-general structure. Collectively, these findings establish the Polish PFS as a reliable, culturally appropriate instrument for tracking the temporal dynamics of optimal experience and illustrate how repeated measurement coupled with multilevel modelling can advance research on flow.

## Introduction

### 1. Flow concept: Past and current understanding

The concept of flow, often described as being "in the zone," refers to a state of optimal psychological functioning in which individuals experience complete absorption in an activity, a sense of effortless control, and intrinsic satisfaction [1]. Introduced by Mihaly Csikszentmihalyi, flow theory has become a cornerstone in positive

**Data availability statement:** The complete dataset and all R scripts required to reproduce the analyses reported in this article are deposited on the Open Science Framework (OSF) and can be accessed via the following link: https://osf.io/bcwk7/overview.

**Funding:** The study presented in the article was conducted as a component of a broader project funded by the Polish National Science Centre (Narodowe Centrum Nauki, NCN). The grant (no: 2024/08/X/HS6/00465) was awarded to M.W. under the MINIATURA 8 funding scheme. Funder websites: https://www.ncn.gov.pl/en https://www.ncn.gov.pl/konkursy/wyniki/miniatura8The funder had no role in study design, data collection and analysis, decision to publish, or preparation of the manuscript.

**Competing interests:** The authors have declared that no competing interests exist. There are no relevant financial or non-financial competing interests to report. Specifically, the authors report no affiliations, memberships, funding sources, financial holdings, advisory roles, or personal relationships that could be perceived as potential sources of conflict or that might influence the work reported in this manuscript. The study was conducted independently of any commercial, political, or institutional interests.

psychology, emphasizing the conditions and experiences that foster human flourishing [2]. Initially, flow was defined as a balance between challenge and skill – an alignment that enables deep engagement and high performance. This early framework was grounded in qualitative studies and interviews, offering rich narratives from individuals who achieved flow in various contexts such as art, sports, and work [2].

The original flow model proposed nine dimensions: clear goals, immediate feedback, challenge-skill balance, action-awareness merging, focused concentration, a sense of control, time transformation, loss of self-consciousness, and intrinsic motivation [3–5]. While these dimensions captured the complexity of the flow experience, subsequent research revealed limitations in the model. Critics pointed out definitional inconsistencies and overlaps between dimensions, particularly the ambiguous distinction between antecedents (e.g., clear goals) and experiential states (e.g., intrinsic motivation) (e.g., [6–8]). In response, recent work has moved toward more parsimonious frameworks that aim to distill the core experiential dimensions of flow. Within contemporary psychological literature, two prominent research groups—led respectively by Peifer [8,9] and Norsworthy [1,10]—have proposed influential, though somewhat divergent, conceptual frameworks to better capture the essential components of flow.

Peifer and colleagues' [9] recent review proposed a three-component model of flow, consisting of: (a) perceived challenge-skill balance, (b) absorption, and (c) enjoyment. In this framework, perceived challenge-skill balance reflects an individual's subjective evaluation of their abilities relative to task demands; absorption denotes deep cognitive engagement and focused attention; and enjoyment refers to the intrinsically rewarding emotional experience associated with flow [9]. While this model offers an intuitively appealing approach, it is not without limitations. First, the notion of perceived challenge–skill balance is debated and ambiguously framed in older accounts: in classic channel/quadrant approaches it is treated alternately as a precondition for flow and as a defining attribute of the state, leading to inconsistent operationalizations and interpretations (cf. [8,11]). Second, since enjoyment reflects a subjective, retrospective evaluation of the activity, rather than a reflection of the experience during the activity itself, treating it as a core flow dimension raises concerns about potential bias [1].

In contrast, the framework developed by Norsworthy et al. [1,10] was explicitly designed to address longstanding critiques within flow research, such as construct ambiguity, definitional inconsistencies, and the frequent conflation of antecedents with core experiential components [6]. Based on an extensive scoping review of interdisciplinary literature on flow, Norsworthy et al. [1,10] identified three distinct yet interconnected core flow dimensions: (a) absorption, (b) effortless control, and (c) intrinsic reward. Absorption in their model closely mirrors Peifer's conceptualization, referring to deep attentional involvement and a merging of action and awareness, characterized neuropsychologically by a reduction in self-referential processing [1,12].

The second dimension, effortless control, clearly distinguishes Norsworthy's model from Peifer's emphasis on perceived challenge-skill balance. Effortless control refers to the subjective experience of fluid and effortless task execution accompanied by a high sense of agency and mastery [1]. Crucially, this conceptualization intentionally

separates the experiential state of effortless action from its antecedent conditions (e.g., appropriate task difficulty and skill level), thereby addressing a central critique in the field [6]. Moreover, effortless control can emerge even in contexts that do not involve achievement-oriented tasks (e.g., engrossing conversations), where a challenge-skill balance may not be explicitly perceived [10].

The third dimension, intrinsic reward, captures the immediate hedonic reward and inherent satisfaction derived from performing the task itself. Norsworthy et al. [10] purposefully use the term intrinsic reward rather than enjoyment [8], as it more directly corresponds with the neurophysiological correlates of flow, such as dopaminergic activation [13,14]. Unlike enjoyment which involves a retrospective evaluation of an activity, intrinsic reward—as core flow dimension – is experienced in the moment and can be potentially measured as the task is being performed, reducing retrospective bias and allowing for more accurate empirical assessment [1,10,15].

In summary, while both frameworks share important commonalities—most notably, the recognition of absorption as a central flow dimension—they diverge significantly in conceptual precision and empirical applicability. Peifer and colleagues' model, though heuristically valuable, continues to employ constructs (such as perceived challenge-skill balance and enjoyment) which conflate antecedents and outcomes with the flow experience itself, complicating construct validity and operational clarity. In contrast, Norsworthy and colleagues' three-dimensional framework—comprising absorption, effortless control, and intrinsic reward – directly responds to previous critiques by offering greater definitional clarity, theoretical coherence, and measurement objectivity.

## 2. Self-Report Instruments for Flow Measurement

Self-report instruments remain the most commonly used and essential tools for measuring flow, offering accessible, versatile, and cost-effective ways to assess subjective experiences across wide range of contexts. As theoretical understandings of flow have advanced, measurement approaches have likewise evolved to better capture the complex and dynamic nature of the phenomenon. However, a critical consideration in selecting and interpreting these instruments involves understanding whether flow is conceptualized as a state-like experience (fluctuating within individuals across contexts or situations) or as a trait-like attribute (stable differences between individuals) [16].

Initial efforts to measure flow, such as Csikszentmihalyi's Flow Questionnaire (FQ), relied on open-ended narratives, encouraging participants to qualitatively describe their optimal experiences [4]. Although valuable for exploratory research, the qualitative nature of these early instruments posed significant limitations for standardization and comparability, particularly in quantitative studies aiming at distinguishing situational variability from stable individual traits.

Subsequent instruments, such as the Flow State Scale (FSS) and its revised version, the FSS-2, were developed to operationalize Csikszentmihalyi's original nine-dimensional model [17]. These tools employ structured Likert-type scales to quantify experiences across dimensions like action-awareness merging and intrinsic motivation [1,4]. Although widely used, the FSS instruments have been criticized for conflating antecedent conditions (e.g., perceived challenge-skill balance) with flow experience and its outcomes (e.g., increase in intrinsic motivation or enjoyment). This conceptual overlap complicates interpretations, particularly when attempting to determine whether measured differences reflect situational states or stable individual traits [4,6,9].

Addressing this critical ambiguity, recent advancements in flow measurement have emphasized a clearer distinction of state-like experiences from trait-like attributes. The Flow Short Scale (FSS), for example, simplifies assessment by focusing explicitly on state-relevant aspects such as absorption and perceived fluency of action, while deliberately excluding antecedent conditions. Its brevity and clarity make it highly effective in repeated-measure designs and studies aiming to capture within-person variability [5,18].

A further methodological advancement is the development of the Psychological Flow Scale (PFS) by Norsworthy et al. [10]. The PFS is designed to assess the core experiential dimensions of flow—absorption, effortless control, and intrinsic reward—while deliberately excluding antecedents or contextual factors. This targeted approach facilitates analyses aimed

at disentangling within-person (state-like) variability from between-person (trait-like) differences. Empirical support from multilevel confirmatory factor analysis (CFA) studies underscores the importance of such an approach, highlights the value of this distinction, demonstrating that using trait-like and state-like measures can lead to fundamentally different, and sometimes even opposing, interpretations [19,20]. For instance, individuals may show significant moment-to-moment variation (within-person) in their flow experiences across contexts or time, while also displaying relatively stable (between-person) differences that reflect personality or trait-like tendencies.

Given the state-trait distinction, research designs that incorporate multilevel analyses (ML-CFA) are crucial for accurately capturing and interpreting flow. These analytical approaches enable researchers to separate within-person (situational flow as a transient state) from stable between-person differences (flow as a personality trait). This methodological rigor ensures more precise conclusions about the nature of flow, aligning with classical definitions emphasizing flow primarily as a state that emerges under specific contextual conditions [2].

In conclusion, self-report measures of flow should explicitly clarify whether the construct being assessed is conceptualized as state-like or trait-like. Instruments like the PFS offer promising methodological clarity by supporting research designs that capture within-person variability, thereby promoting a more accurate understanding of flow as a dynamic psychological state. This approach is consistent with contemporary calls for greater precision in the definition and measurement psychological constructs [10,19,20], ultimately enhancing both theoretical clarity and practical application of flow research.

## 3. Current study

Empirical studies exploring flow as a psychological state have frequently employed the Experience Sampling Method (ESM) due to its sensitivity in capturing moment-to-moment fluctuations within everyday contexts. Originally developed by Csikszentmihalyi and colleagues, ESM involves repeated, real-time assessments, allowing researchers to track psychological states as they naturally unfold across diverse situations [21]. This methodological flexibility has enabled investigations of flow across wide range domains, including occupational settings [22], educational environments [21], healthcare contexts [23], and even daily leisure activities [24]. Moreover, recent studies employing ESM have revealed complex and nonlinear patterns of flow over time [25–27].

Acknowledging these methodological insights, the present study emphasizes the importance of strict experimental control to reliably induce and measure flow as a psychological state. Specifically, we propose to adapt the PFS [10] for Polish-speaking populations through a highly controlled, standardized laboratory experiment. Participants will engage in a visuomotor tracking task meticulously designed to dynamically match challenge levels with participant skills—conditions considered essential for eliciting flow. Such controlled task manipulation directly addresses the methodological variability of previous ESM research, allowing more robust causal interpretations of flow as a state-like phenomenon.

In addition to validating the Polish adaptation of the PFS, the present study sought to empirically examine the latent structure of flow. Although flow is widely conceptualized as a multidimensional state comprising absorption, control, and intrinsic reward [10], it remains an open question whether these dimensions function independently or reflect a higher-order, unified construct. To address this, we fit a sequence of single- and two-level models that explicitly separated within-from between-person variance. At each level, we compared (a) a three-factor solution with correlated but distinct facets and (b) a hierarchical solution in which a second-order Flow factor accounts for their covariation. This multilevel strategy avoids conflating momentary fluctuations with stable individual differences and provides a direct test of whether the higher-order organization of flow generalizes across levels.

## 4. Method

**Participants.** The recruitment period for this study started on 12 November 2024 and ended on 1 February 2025. The initial sample consisted of 151 Polish-speaking individuals (101 women, 50 men). Eleven participants were excluded from the analyses because they did not complete the full experimental session. The final analytic sample comprised 140

participants (96 women, 44 men). The overall age was M = 21.70 years (SD = 2.79); women: M = 21.20 (SD = 2.28), men: M = 22.80 (SD = 3.46).

The study was conducted in accordance with the ethical standards of the World Medical Association's *Declaration of Helsinki* for research involving human participants. The protocol was approved by the Research Ethics Committee of the Institute of Psychology, John Paul II Catholic University of Lublin (Approval No. KEBN_26/2024). All participants provided written informed consent and were compensated PLN 132 per hour for their participation.

**Apparatus and materials.** The experiment utilized a pursuit-tracking task designed to elicit flow states through a dynamic and visually engaging environment (Fig 1).

The task featured a moving object following a trajectory generated by the Lorenz system equations, creating deterministic chaotic dynamics:

$$\frac{dx}{dt} = \sigma(y - x)$$

$$\frac{dy}{dt} = x(\rho - z) - y$$

$$\frac{dz}{dt} = xy - \beta z$$

The parameters used in the equation were : $\sigma = 5$, $\rho = 29.9990234375$, $\beta = 8/3$.

Lorenz trajectory (x vs y)

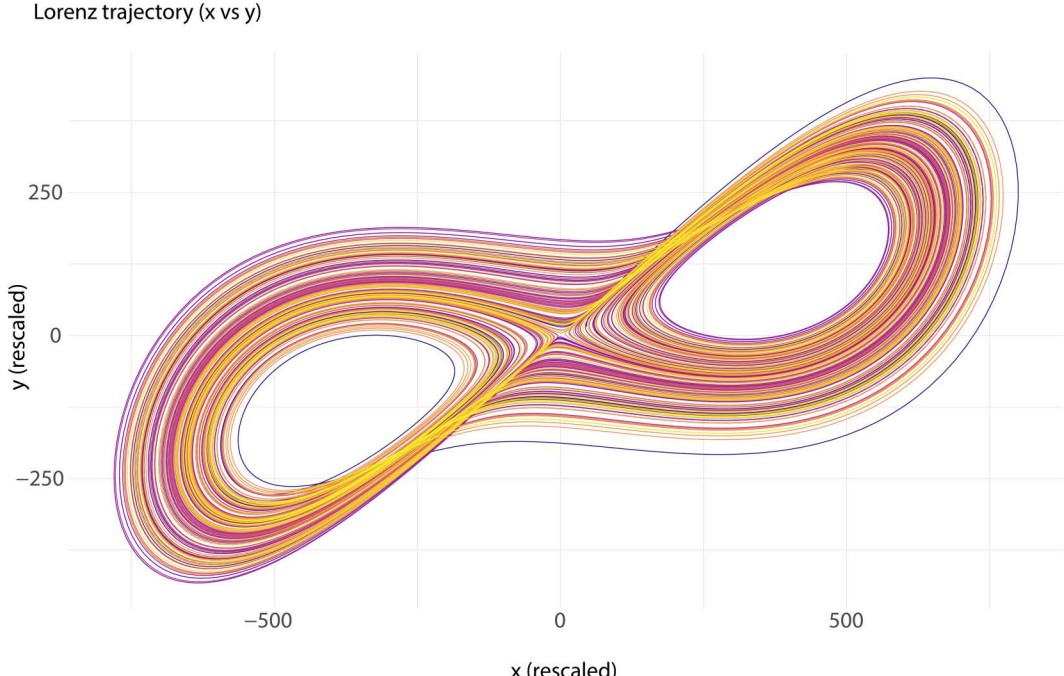

**Fig 1. Deterministically chaotic Lorenz trajectory (rescaled x × y projection) that served as the target path in the pursuit-tracking task.**

Although the target trajectory generated by the Lorenz's system is fully deterministic, it appears unpredictable to the participant, thus giving the impression of randomness.

The moving target object was rendered as a red disc, while participants controlled a cursor (a black circle) using a high-precision Logitech M705 mouse. Task difficulty was dynamically adjusted by modifying the cursor's diameter to maintain an optimal challenge-skill balance (see Procedure).

Although the current study focused primarily on self-report measures of flow, a multimethod design was employed to capture this complex experiential state more comprehensively. Self-reports are widely used and theoretically grounded in flow research, offering valuable first-person insights into subjective experience [28]. Nevertheless, flow is increasingly conceptualized as involving not only subjective absorption but also physiological and behavioral changes. In line with this perspective, the present experiment included a battery of objective measures—continuous mouse-movement tracking, facial electromyography (EMG), electrodermal activity (EDA), eye-tracking metrics, and heart-rate variability (HRV)—collected while participants engaged with the task. These physiological/behavioral recordings were acquired to enable future multimethod analyses but are outside the scope of the current article and are therefore not discussed here; they will be reported in a separate paper. In this article, we used one of the behavioral measures—task accuracy (correctness)—as a variable to test the convergent validity of the PFS. By task accuracy, we refer to the percentage of time during which the distance between the cursor and the target remained within the individually determined threshold, established for each participant during the staircase calibration procedure described in detail below.

Self-report measures provided subjective insights into participants' flow experiences. Momentary flow was assessed with the Psychological Flow Scale (PFS; [10]), which conceptualizes flow as three facets—absorption ("deep, undistracted attention, with a merging of action and awareness"), effortless control ("a heightened sense of control in which the task feels unusually effortless and fluid"), and intrinsic reward ("an intrinsically rewarding experience marked by positive valence and optimal arousal"). The PFS comprises nine items (three per facet) rated on a 7-point Likert scale (1 = Strongly disagree to 7 = Strongly agree), with higher scores indicating more intense flow. In its original validation, the total scale showed excellent internal consistency (α = .85), with subscale alphas ranging from .82 to .93. Beyond internal consistency, the original PFS accumulated multi-source validity evidence—expert review and target-user feedback (content validity), iterative item reduction, EFA/CFA supporting a three-facet structure nested under a higher-order Flow factor, and convergent/discriminant associations with related constructs—consistent with best-practice guidance for modern test validation [10,29,30].

For the present study, the English-language PFS was translated into Polish following best practices for cross-cultural adaptation [31]. Three bilingual researchers produced independent forward translations (see S1 Table for PFS original and Polish items); versions were reconciled to a consensus draft and checked against the source to resolve semantic and conceptual discrepancies, with wording finalized by agreement. This process aligns with contemporary recommendations for ensuring psychometric robustness and measurement invariance across populations [32]. To preserve direct comparability with the source instrument and enable future multi-group invariance tests, we deliberately retained the final nine items of the original PFS rather than re-selecting from the broader content-validated pool. Notably, PFS items target generic, activity-agnostic aspects of momentary flow and contain no culture-specific idioms; the excellent fit indices obtained here further support the adequacy of the Polish phrasing.

In addition to the PFS, participants completed the Polish version of the Flow Short Scale (FSS; [18]; Polish adaptation [33]), a widely used instrument for assessing state-level flow. The scale comprises ten items presented after a defined task segment, of which eight are scored. These items are commonly grouped into two dimensions reflecting key components of the flow experience. The first facet, fluency, captures the sense that one's actions unfold smoothly and effortlessly. Example items include: *"Nie zauważyłem(-am), jak upłynął czas"* ("I did not notice time passing") and *"Miałem(-am) poczucie, że jestem kompetentny(-a)"* ("I felt that I was competent"). The second facet, absorption, relates to focused attention and deep involvement. Example items include: *"Byłem(-am) całkowicie skoncentrowany(-a) na tym, co*

*robiłem(-am)"* ("I was completely focused on what I was doing") and *"Nie miałem(-am) żadnych trudności z koncentracją"* ("I had no difficulty concentrating"). Items were rated on a 7-point Likert-type scale ranging from 1 ("not at all") to 7 ("very much"), with higher scores indicating a more intense flow experience. In the current sample, the fluency subscale (items 2, 4, 7, 8, and 9) demonstrated marginal internal consistency (Cronbach's α = .74). In contrast, the absorption subscale (items 3, 6, and 10) exhibited extremely low reliability (α = .19). Given this, we did not analyze the two-factor structure further nor interpret subscale-level effects. Instead, we relied on the overall flow-intensity score, computed as the mean of all eight scored items. This composite demonstrated marginal internal consistency (α = .72) and was used in all subsequent analyses.

Dispositional flow proneness was measured using the 13-item General Flow Proneness Scale (GFPS; [34]), designed to capture an individual's general tendency to experience flow across diverse activities [35]. Example items include: *" Lubię wymagające zadania/aktywności, które wymagają dużego skupienia."* ("I enjoy challenging tasks/activities that require a lot of focus") and *" Gdy skupiam się na zadaniu/aktywności, mam skłonność do szybkiego zapominania o otoczeniu (innych ludziach, czasie i miejscu)."* ("When I am focused on a task/activity, I quickly tend to forget my surroundings (other people, time, and place)"). Respondents indicate agreement on a 5-point Likert scale (1 = Strongly Disagree to 5 = Strongly Agree). Five items (6, 7, 8, 11, and 12) are reverse-keyed to control for acquiescence and were recoded prior to analysis. In the present sample, the GFPS total score demonstrated excellent internal consistency (Cronbach's α = .97), supporting its reliability as a measure of trait flow proneness.

Objective behavioral indices (mouse tracking). In parallel with self-reports, we recorded mouse trajectories during the pursuit-tracking task and derived standard kinematic/accuracy indices (e.g., correctness) using conventional preprocessing and time normalization. These measures served as an exploratory check of response-process validity for the PFS.

## Procedure

The experiment was preregistered on OSF (https://osf.io/9tnmr) and consisted of three sequential phases designed to optimize engagement and measurement precision. First, a 5-minute practice session allowed participants to familiarize themselves with the pursuit-tracking task and adapt to the dynamic, chaotic trajectory generated by the Lorenz system. This introductory phase aimed to reduce novelty effects and ensure that participants understood the task mechanics before formal data collection commenced.

Next, a skill calibration phase was used to determine individual motor proficiency. The skill-estimation phase consisted of 30 consecutive 10-s trials using a 1-up/1-down staircase to adapt task difficulty to each participant's performance. At trial 1, both the cursor and the tracked disc started at the screen center. Trials proceeded without breaks; the disc's trajectory was continuous across trials and generated by the Lorenz equations. Difficulty was manipulated by changing the diameter of the tracking circle (Δd), with the staircase targeting a success probability of 0.50 (i.e., trial "success" coded when the target was inside the circle for ≥ 5 s out of 10 s). After each trial, the next trial became harder if the current trial was successful and easier if it was unsuccessful (1-up/1-down). Reversals were defined as changes in the adjustment direction. Step sizes followed a three-stage schedule: Δd = 0.8 from the start until the second reversal, then Δd = 0.4 until the fourth reversal, and Δd = 0.2 thereafter. The initial circle diameter was 2.0 (bounded to 0.4–5.0). The final converged diameter was taken as the skill parameter (larger values = lower skill; smaller values = higher skill) and was held constant during the 20 experimental trials. The mean skill score was M = 2.96 (SD = 0.69).

In the main phase of the experiment, participants completed 20 full-length trials, each lasting 3 minutes. The difficulty level established individually for each participant during the calibration phase was held constant throughout this stage. Following each trial, participants completed the PFS to self-report their subjective flow experience.

Full details of the experimental protocol, including screen layouts, timing parameters, and example trial scripts, are available in the preregistration document: https://osf.io/9tnmr.

## Statistical analyses

All analyses were conducted in R 4.3 [36]. Data were pre-processed and reshaped with dplyr and tidyr [37, 38]. We first evaluated reliability using a generalizability-theory multilevel approach tailored to the data's three-level structure (items within trials within participants), yielding indices that capture generalizability across items and occasions, between-person consistency, and sensitivity to within-person fluctuations [39]. Factorial validity was then examined via a step-wise MCFA in lavaan [40], progressing from a single-level three-factor model, through a hierarchical (second-order) specification, to a two-level model that separates within- and between-person variance; the final solution featured a hierarchical (second-order) structure at both the within-person (Level 1) and between-person (Level 2) levels, and included theoretically justified residual covariances between selected items at the between-person level. To cross-validate the structure with a theory-agnostic method, we ran hierarchical Exploratory Graph Analysis on item scores averaged across trials (EGAnet; [41]). Temporal dynamics were summarized by computing trial-level facet/Flow scores and visualizing them with ggplot2 [42], followed by mixed-effects models with random intercepts for participants to test linear change over time (lme4; [43]). Convergent validity was assessed by correlating (a) PFS scores from the final trial with FSS (state-level) and (b) aggregated PFS scores (20-trial means) with GFPS (trait-level); we additionally computed trial-by-trial correlations between GFPS and PFS facets to probe the stability and selectivity of trait–state associations across the session.

To determine whether our study had sufficient power to test the most complex model, we conducted a Monte Carlo simulation procedure [44]. Based on a multilevel model with nine observed variables (items), three first-order factors, and one second-order factor, we generated 100 artificial datasets including between and within-person effects. We set factor loadings at .7. Using the results from the 100 datasets, we examined: (1) the percentage of significant (alpha = .05) factor loadings associated with first- and second-order latent variables, (2) the percentage of estimation errors (i.e., Heywood cases), (3) the percentage of simulations in which the model did not converge, and (4) the percentage of simulations with optimal fit indices, i.e., RMSEA < .08 and CFI ≥ .95. We performed the Monte Carlo analysis for sample sizes ranging from 20 to 160 participants (step = 20). Each simulation assumed 20 trials per person, resulting in total sample sizes from 400 to 3,200 observations. The results of the power analysis for each tested sample size are presented in the Supplemental Materials (see S1 Fig for Monte Carlo power analysis results). The R script used for these simulations is available on OSF (link provided). For the sample size equal to that of our empirical study (N = 140), the percentage of significant factor loadings at the between-person level was 94% for first-order loadings and 87% for second-order loadings; for the within-person level, 100% of loadings were significant. We did not observe any simulations with Heywood cases or model non-convergence. In every simulation, the model met the fit criteria (RMSEA < .08 and CFI ≥ .95). Our Monte Carlo analysis therefore indicates that the number of participants in our study was sufficient to achieve power at least $\beta = .87.5$. Analyses

To assess the internal consistency and generalizability of the Polish adaptation of the PFS, a multilevel reliability analysis was conducted using generalizability theory [45], implemented via the multilevel.reliability() function from the psych package [39]. The data followed a three-level structure: items (9) nested within measurement occasions (20 trials), nested within participants (N = 140).

The results revealed excellent generalizability of average scores across all items and time points (RkF = 1.00), supporting the high reliability of the overall flow score when aggregated. Similarly, the generalizability of average scores across time (RkR = .97) and the consistency of between-person differences over time (RkRn = .97) were both very high, reflecting strong stability in individuals' relative standing across repeated measurements.

The generalizability of change (Rc = .88) was also high, suggesting that the PFS is sensitive to within-person fluctuations in flow across repeated trials. The generalizability of within-person variability across items (Rcn = .67) and the reliability of single time-point estimates (R1R = .65) were somewhat lower, indicating moderate reliability when examining flow states at isolated time points.

The variance decomposition further illustrated that the largest share of variance was attributed to person-level differences (31%), residual error (18%), and item-by-person interactions (20%; i.e., stable person-specific tendencies to endorse certain items more or less across trials, beyond one's overall level). These findings align with the theoretical complexity of flow as a dynamic and context-sensitive psychological state, shaped both by individual differences and situational variability. Results collectively support the appropriateness of a multilevel analytic approach, in which flow is modeled both as a dynamic state fluctuating within individuals and as a relatively stable construct differentiating between individuals.

To evaluate the factorial structure of the PFS, a series of CFAs were conducted progressively increasing the model complexity, ultimately employing a multilevel structural equation modeling (SEM) framework appropriate for repeated measures nested within individuals (20 trials × 140 participants; Table 1).

We evaluated the factor structure of the PFS in six steps, moving from single-level CFA to a fully multilevel, hierarchical specification appropriate for 20 repeated trials nested within 140 participants. As a baseline, Model 1 treated Absorption, Effortless Control, and Intrinsic Reward as three correlated first-order factors at a single level and already fit well (e.g., $\chi^2(24) = 108.86$, CFI = .988, RMSEA = .067). Model 2 re-expressed the same covariance by introducing a second-order Flow factor—yielding the expected, essentially identical fit—thereby establishing a hierarchical interpretation at the aggregate level.

To separate momentary (within-person) from stable (between-person) variance, Model 3 extended the structure to two levels, specifying the same three correlated factors within and between persons. Fit improved and was very good overall, but a small Heywood case on the *between* side suggested minor local dependence among indicators. We addressed this in Model 4 by allowing two theoretically justified residual covariances at the between-person level (paralleled item content), which stabilized residuals and maintained excellent global fit.

Next, we asked whether Flow organizes item covariation not only across people but also across occasions. Model 5 therefore added a second-order Flow factor at the *within-person* level (retaining Model-4 structure between persons). This preserved the strong fit and showed high, interpretable second-order loadings. Finally, Model 6—our final model—implemented the hierarchical Flow factor on both levels, aligning most closely with theory (a unifying Flow construct that governs trial-to-trial fluctuations and between-person differences). Its fit was virtually indistinguishable from Models 4–5 (e.g., $\chi^2(46) = 345.35$, CFI = .981, RMSEA = .048; SRMR_within =.039; SRMR_between =.037), but it provides the clearest, most parsimonious account of the data-generating process.

Based on theoretical grounding, empirical fit, and parsimony, Model 6—a two-level hierarchical CFA with a second-order Flow factors both at the within-person level and at the between-person level—was selected as the final model (Fig 2).

Model 6 exhibits excellent fit with no estimation irregularities and is supported by both the within-person dynamics and the between-person stability observed in the data, consistent with earlier findings (e.g., [10]).

**Table 1. Model fit indices for six CFA specifications.**

| Model | Structure (levels) | $\chi^2$ (df) | CFI | TLI | RMSEA [90% CI] | SRMR (within/ between) |
|---|---|---|---|---|---|---|
| 1. Three correlated factors | single-level | 108.86 (24)† | .988‡ | .981‡ | .036 [.032,.039]† | .031 (—) |
| 2. Second-order Flow (within) | single-level | 108.86 (24)† | .988‡ | .981‡ | .036 [.032,.039]† | .031 (—) |
| 3. 3 correlated factors (within & between) | two-level | 348.47 (48) | .981 | .972 | .047 [.043,.052] | .039/.036 |
| 4. Two-level + L2 residual covariances | two-level | 345.35 (46) | .981 | .971 | .048 [.044,.053] | .039/.037 |
| 5. Two-level + second-order Flow at L1 | two-level | 345.35 (46) | .981 | .971 | .048 [.044,.053] | .039/.037 |
| 6. Two-level + second-order Flow at L1 & L2 | two-level | 345.35 (46) | .981 | .971 | .048 [.044,.053] | .039/.037 |

*Notes* All models were estimated on 2,800 observations clustered in 140 participants. For Models 1–2 (single-level), we report Satorra–Bentler scaled $\chi^2$ and the corresponding "robust" fit indices; RMSEA is the scaled value from lavaan's robust output.† Robust CFI/TLI are shown in the CFI/TLI columns.‡ Two-level models (3–6) used ML/EM and report conventional (non-robust) indices from lavaan.

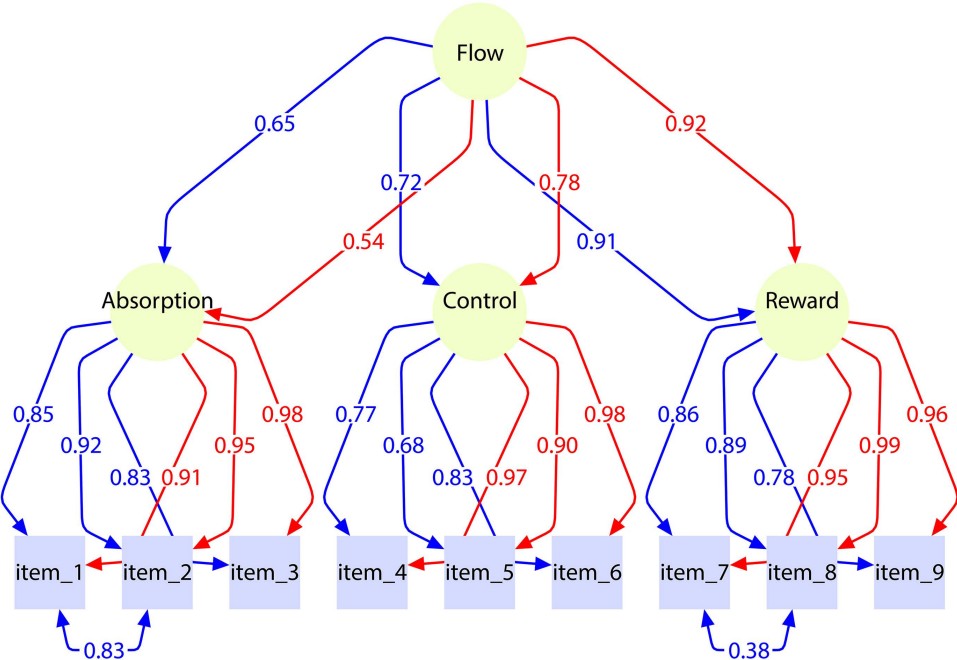

**Fig 2. Final multilevel confirmatory factor model of the Polish Psychological Flow Scale (PFS; N = 140).** Blue arrows represent between-person (Level 2) factor loadings, whereas red arrows represent within-person (Level 1) factor loadings.

The following section presents the distribution of flow-related variables across individual trials, further disaggregated by specific subscales (Fig 3).

Visual inspection suggests only small group-level changes across trials. Mixed-effects modeling nevertheless indicates a modest linear decline in overall flow ($\beta = -0.181$ per 1-SD change in Trial; $t(2659) = -12.25$, $p < .001$), corresponding to $\approx$ $-0.031$ points per trial and $\approx$ $-0.60$ from Trial 1–20 (scale 1–7).

Decomposition (Fig 4) shows the same downward drift for Absorption ($\beta = -0.316$; $t(2659) = -16.86$, $p < .001$; $\approx -0.055$ per trial; $\approx -1.04$ total) and Intrinsic Reward ($\beta = -0.233$; $t(2659) = -13.19$, $p < .001$; $\approx -0.040$ per trial; $\approx -0.77$ total), whereas Effortless Control shows no trend ($\beta = +0.006$; $t(2659) = 0.31$, $p = .760$; $\approx +0.001$ per trial; $\approx +0.02$ total).

To further corroborate the factorial validity of the selected multilevel hierarchical CFA model, we employed a complementary data-driven approach using Hierarchical Exploratory Graph Analysis (hierEGA) [46–48]. This technique allows for the identification of both lower-order and higher-order latent structures in multivariate data based on network modeling principles, offering a theory-agnostic confirmation of emergent dimensions. The EGA was conducted on aggregated data, where PFS item scores were averaged across all 20 trials for each participant (N = 140; Fig 5).

The resulting network analysis revealed a clear and stable three-cluster solution at the lower level, with items 1–3 loading together (Absorption), items 4–6 clustering (Effortless Control), and items 7–9 forming a third group (Intrinsic Reward). These clusters exactly mirrored the hypothesized structure based on Norsworthy et al.'s [10] model, and aligned with the first-order factors specified in the multilevel CFA.

At the higher-order level, the three dimensions converged into a single overarching factor. The TEFI (Total Entropy Fit Index) is an information-theoretic fit measure based on Von Neumann entropy that quantifies how well a proposed dimensional structure reduces disorder in the item-correlation network; lower (more negative) values indicate better fit. The generalized TEFI (gTEFI) extends this logic to hierarchical/bifactor comparisons. In our data, the higher-order structure fit better (TEFI = −9.98; gTEFI = −12.74) than the correlated lower-order model (TEFI =

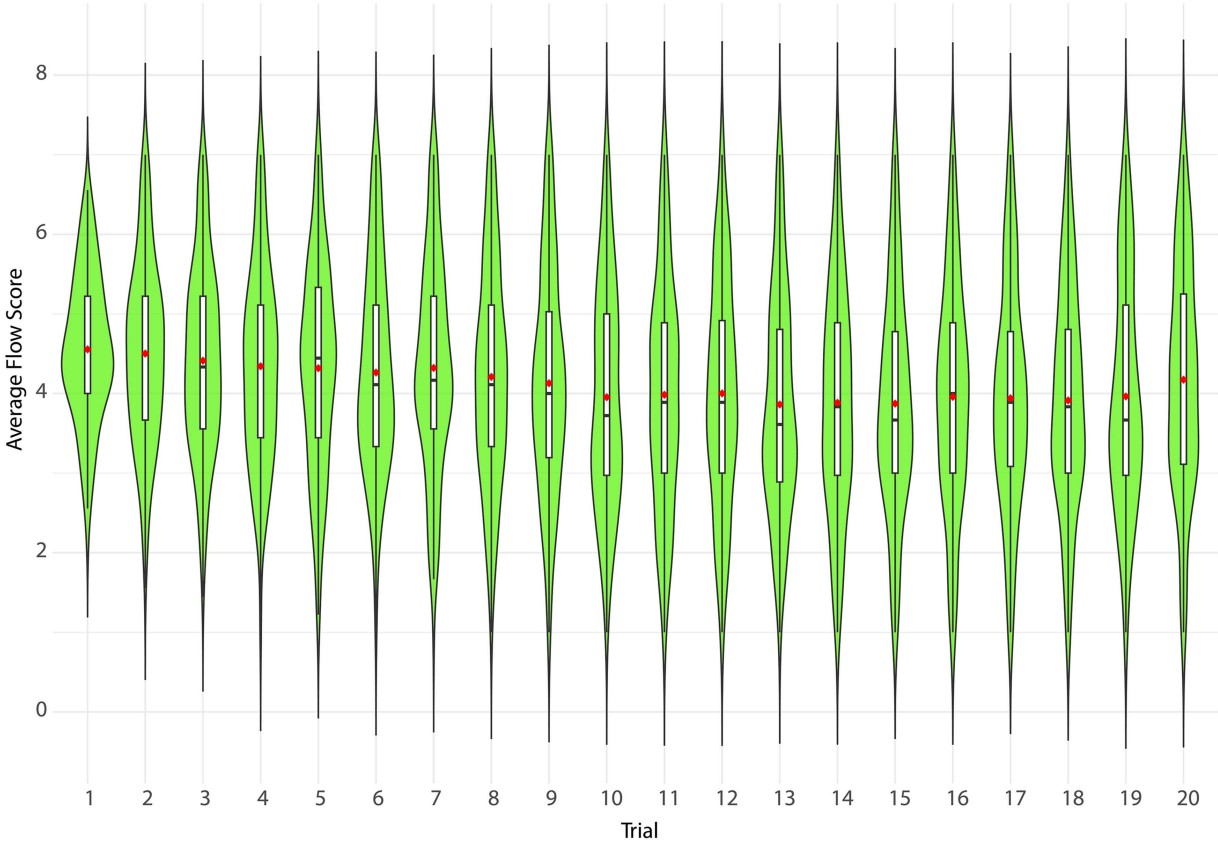

**Fig 3. Stability of overall flow across twenty consecutive trials (N = 140).** Each green violin represents the distribution of average flow scores in a single trial (wider sections indicate a higher density of observations at that score). The white box inside each violin represents the interquartile range (25th–75th percentile), the black horizontal line within the box marks the median, and the red dot indicates the mean average flow score for that trial. The vertical black lines ("whiskers") extend to the most extreme values within 1.5 × IQR.

−2.77), suggesting that a bifactor or second-order solution provides a superior account of the covariance among items.

This independent, data-driven result provides compelling converging evidence for the hierarchical nature of flow as a psychological construct. Specifically, it supports the notion that Absorption, Effortless Control, and Intrinsic Reward operate as interrelated subcomponents of a broader experiential flow state, rather than as entirely independent constructs.

To assess convergent validity, correlation analyses were conducted using two established instruments and one task-accuracy index. The first analysis involved comparing the adapted PFS [10], measured after the final (20th) trial, with the Flow Short Scale (FSS), also administered after the 20th trial (Table 2).

The convergent-validity analyses comparing the PFS (trial 20) with the Flow Short Scale (FSS) showed positive associations across all pairings. Correlations with the FSS Fluency subscale were small-to-moderate overall (r ≈ .23–.38), with the largest coefficients for PFS Intrinsic Reward and Effortless Control. By contrast, correlations involving the FSS Absorption subscale were uniformly small (r ≈ .19–.26). Importantly, in this dataset the FSS Absorption subscale displayed very low internal consistency (α ≈ .20), which likely attenuates its observed correlations; these coefficients should therefore be treated as lower-bound, descriptive estimates and interpreted with caution. For transparency we report both FSS subscales in Table 2, but—given this reliability limitation—we rely on the FSS total score (r ≈ .28–.39 with PFS facets) as the primary state-level criterion when summarizing convergent evidence in the main text.

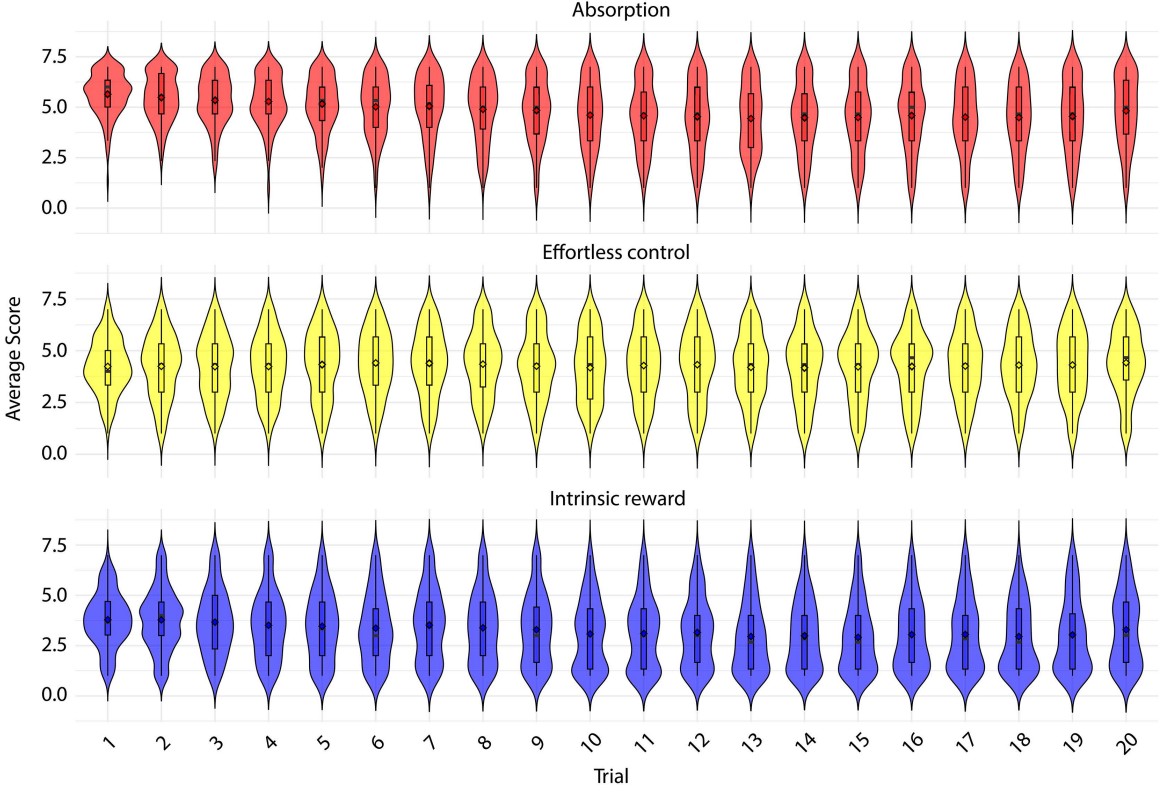

**Fig 4. Facet-level stability: trial-by-trial distributions of Absorption, Effortless Control and Intrinsic Reward (N = 140).** Each coloured violin represents the distribution of average facet scores in a single trial (wider sections indicate a higher density of observations at that score). The box within each violin shows the interquartile range (25th–75th percentile), the horizontal line marks the median, and the whiskers extend to the most extreme values within 1.5 × IQR.

The second analysis tested the relationship between state-level flow experiences and trait-level flow proneness by correlating average PFS scores (across all 20 trials) with the General Flow Proneness Scale (GFPS) (Table 3).

The analysis of convergent validity between the trait measure of flow (GFPS) and aggregated PFS scores (20-trial means) showed small-to-moderate trait–state associations overall, strongest for Intrinsic Reward and Effortless Control. By contrast, the link with Absorption was weak and not statistically significant (r = .13, 95% CI [−.03,.29]).

As shown in Fig 6, trait flow proneness shows low-to-moderate positive associations with Effortless Control and Intrinsic Reward, whereas Absorption is only weakly related.

We also used task-accuracy index (i.e., percentage of time during which the distance between the cursor and the target did not exceed the threshold determined individually for each participant during the staircase calibration procedure; M = .54, SD = .09) to determine convergent validity of PFS. The within-person correlations between task-accuracy and absorption, effortless control, intristic reward, and flow total score were r = .13 (p < .001), r = .32 (p < .001), r = .15 (p < .001), and r = .25 (p < .001), respectively. At the between-person level correlations were as follows: r = .28 (p = .001), r = .26 (p = .002), r = .22 (p = .009), and r = .31 (p < .001), respectively.

## 5. Discussion

The aim of the present study was to validate the Polish adaptation of the PFS and to examine the factorial structure of flow using a tightly controlled experimental paradigm. By employing a multilevel analytical approach, the study was able to

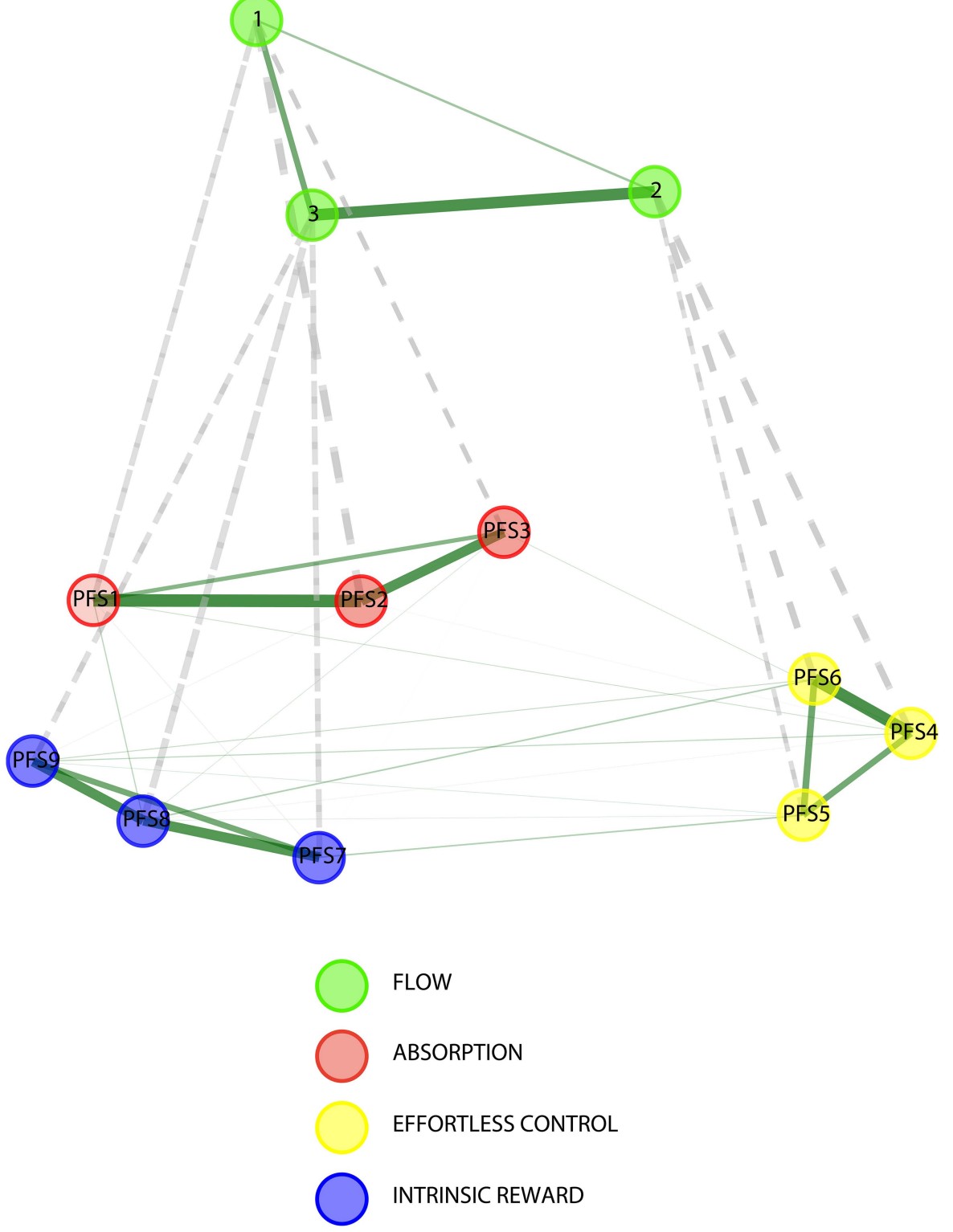

**Fig 5. Hierarchical Exploratory Graph Analysis (hierEGA) of aggregated Psychological Flow Scale item scores (N = 140).** The thickness of the solid lines (edges) reflects the strength of the association between nodes, whereas the dashed grey lines indicate connections between nodes at the higher flow level and nodes at the lower facets level.

**Table 2. Descriptive statistics and convergent-validity correlations (Pearson's *r*) between Polish PFS facets and Flow Short Scale scores obtained after the final trial.**

| Scale/subscale | M | SD | Absorption | Effortless Control | Intrinsic Reward |
|---|---|---|---|---|---|
| Absorption (PFS) | 4.82 | 1.73 | | | |
| Effortless Control (PFS) | 4.42 | 1.60 | .48*** (.34−.60) | | |
| Intrinsic Reward (PFS) | 3.29 | 1.80 | .56*** (.43−.66) | .61*** (.49−.70) | |
| Absorption (FSS) | 4.17 | 1.06 | .23** (.06−.38) | .26** (.10−.41) | .19* (.03−.35) |
| Fluency (FSS) | 4.46 | 1.09 | .23** (.06−.38) | .31*** (.15−.45) | .38*** (.23−.51) |
| Flow Short Scale Total Score | 4.35 | 0.87 | .28*** (.12−.43) | .36*** (.20−.49) | .39*** (.23−.52) |

*Note.* PFS facet scores are trial-20 means on a 1–7 scale: Absorption (PFS) = items 1–3; Effortless Control (PFS) = items 4–6; Intrinsic Reward (PFS) = items 7–9. FSS scores are means on a 1–7 scale: Fluency (FSS) = items 2, 4, 7, 8, 9; Absorption (FSS) = items 3, 6, 10; FSS Total = eight scored items. Entries are Pearson's *r* with two-tailed 95% CIs in parentheses. *N* = 140. Internal consistency (Cronbach's α) in this sample: Fluency (FSS) = .74; Absorption (FSS) = .20; FSS Total = .78. Labels "(PFS)" vs. "(FSS)" distinguish the instruments.

**Table 3. Descriptive statistics and trait–state convergent-validity correlations (Pearson's *r*) between aggregated Polish PFS facets (20-trial means) and the General Flow Proneness Scale.**

| Scale/subscale | M | SD | Absorption | Effortless Control | Intrinsic Reward |
|---|---|---|---|---|---|
| Absorption | 4.84 | 1.25 | | | |
| Effortless Control | 4.28 | 1.29 | .40*** (.25−.53) | | |
| Intrinsic Reward | 3.27 | 1.43 | .49*** (.36−.61) | .70*** (.60−.77) | |
| General Flow Proneness Scale | 3.40 | 0.52 | .13 (−.03−.29) | .20** (.04−.36) | .26*** (.09−.41) |

*Note.* PFS facets are 20-trial means (scale 1–7). GFPS is the item mean (scale 1–5). Correlations are between-person Pearson's r; 95% confidence intervals in parentheses.

distinguish between transient, within-person fluctuations in flow and more stable, between-person differences—an essential distinction given the state-like nature of flow as theorized in recent literature [1,2,20].

The results offer robust support for the conceptualization of flow as a multidimensional psychological state. Analyses confirmed the internal reliability of the PFS, not only in its ability to differentiate between individuals, but also in capturing meaningful within-person variability across trials. These findings align with the assumption that flow is a dynamic experience that emerges and dissipates over time [22,26], and that any valid measure of flow must be sensitive to these moment-to-moment changes [10,15].

Confirmatory factor analyses indicated that the best-fitting and theoretically most coherent solution was a two-level hierarchical model. In this specification, a second-order Flow factor was defined at both the within-person and the between-person levels, capturing the common variance among Absorption, Effortless Control, and Intrinsic Reward at each level while preserving their distinctiveness. This structure mirrors contemporary formulations of flow as a unified state emerging from separable yet interrelated components (e.g., [10]).

Importantly, the second-order specification at both levels converged with excellent fit and yielded interpretable parameters. A higher-order Flow factor at the within-person and between-person levels parsimoniously captured the common

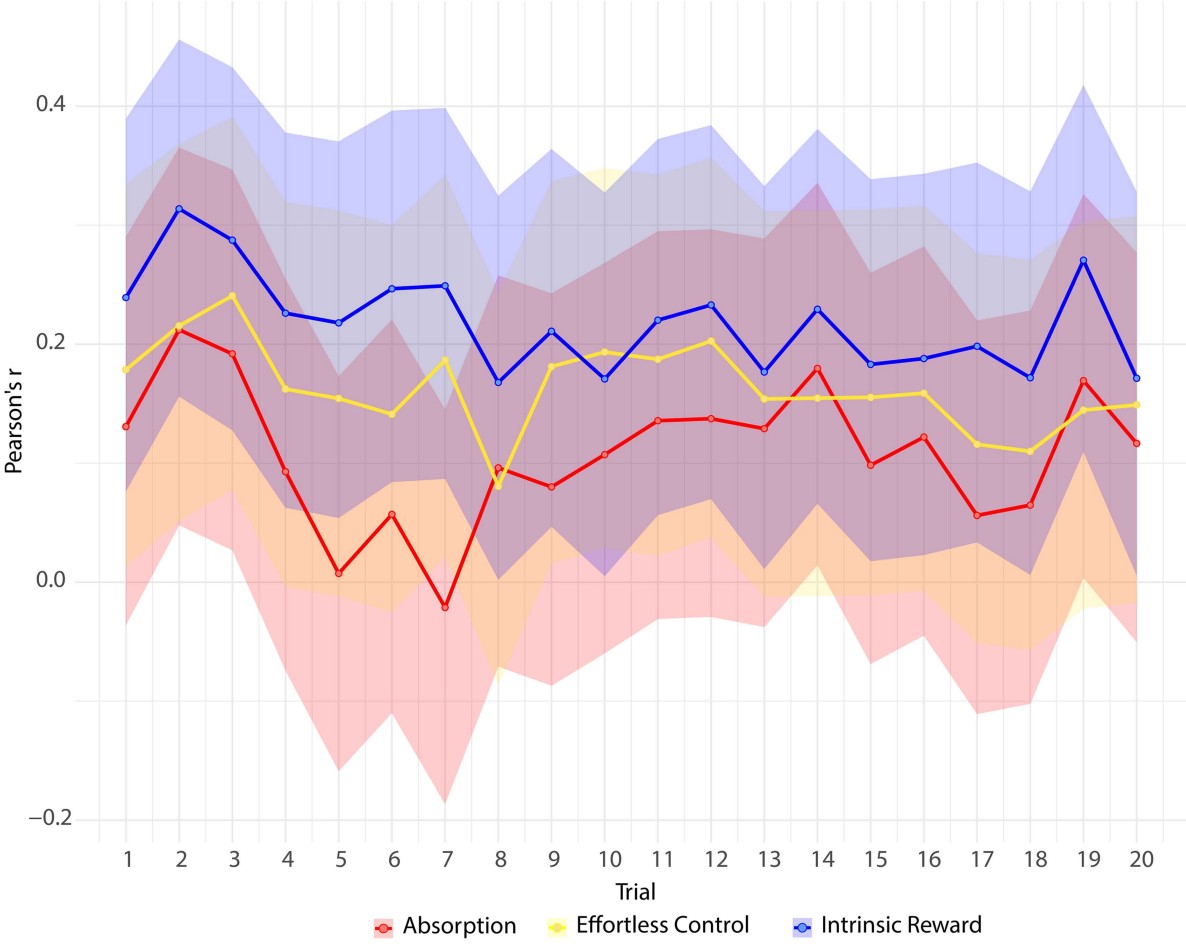

**Fig 6. Trial-by-trial correlations between trait flow proneness and momentary flow components (N = 140).** Solid lines represents fluctuations in Pearson's r coefficients across 20 trials. The shaded ribbons around each line represent 95% confidence intervals for these estimates.

variance among Absorption, Effortless Control, and Intrinsic Reward while preserving their distinctiveness. This pattern indicates that the hierarchical organization of flow is evident not only in moment-to-moment dynamics but also in stable individual differences, consistent with recommendations for modeling higher-order constructs in multilevel SEM [19,49]. Converging evidence came from hierarchical Exploratory Graph Analysis on aggregated item scores, which reproduced a robust three-cluster lower-order structure mapping onto the PFS facets and a single superordinate dimension corresponding to Flow [46,47].

The study also provided supporting evidence for the convergent validity of the PFS. State-level flow experiences captured immediately after task performance were positively associated with another validated state-based measure (FSS; [18]). Also, both PFS subscales and total score significantly and positively correlated with performance measure, i.e., task accuracy. However, when the PFS scores were aggregated across the twenty trials and related to the General Flow Proneness Scale (GFPS), the links were markedly weaker: Effortless Control and Intrinsic Reward showed only modest associations and Absorption was essentially unrelated. These findings only partly support the claim that momentary flow is shaped by enduring individual predispositions [1,50]. Thus, while the PFS clearly captures the construct's dynamic, situational facet, its capacity to reflect more stable trait-like characteristics still requires further investigation.

Several limitations should be acknowledged. First, the study employed a single, tightly controlled pursuit-tracking task, which enhances internal validity but restricts ecological generalizability; everyday flow unfolds in richer and more varied contexts [21,23,24]. Second, the sample was comparatively homogeneous (young, university-educated, and right-handed), limiting representativeness. Third, the evidence is based on self-report measures that, although informative, are vulnerable to cognitive biases and demand characteristics; the concurrently recorded objective indicators (facial EMG, EDA, eye-tracking, HRV) were not analyzed here and therefore do not provide triangulation in the present report [13,51,52]. Fourth, convergent validity with the GFPS was modest—particularly for Absorption—so trait–state correspondence should be interpreted with caution, potentially reflecting a content and timescale mismatch (broad disposition vs. task-bound, momentary absorption) and range restriction under laboratory conditions. Fifth, the FSS Absorption subscale exhibited very low internal consistency ($\alpha \approx .20$), which precludes reliable subscale-level inferences; analyses therefore relied on the FSS total score when benchmarking state flow [18,33]. A further limitation is that we did not evaluate content validity beyond semantic/linguistic equivalence; we did not re-appraise the wider PFS item pool with Polish experts or lay reviewers. This choice preserves cross-study comparability with the original PFS but may have foregone incremental gains in clarity or reliability that alternative Polish phrasings could provide. Next consideration is that some participants (N = 20) exhibited little or no within-person variability across trials. Such profiles are not inherently problematic in multilevel designs and remain informative at the between-person level; our modeling approach explicitly separates within- and between-person variance. Future work may characterize whether low-variance profiles reflect stable, high flow or other response processes (e.g., via behavioral/physiological indices). Finally, the reported trait–state correlations are between-person estimates, and time trends were modeled as linear; within-person coupling and non-linear or person-specific dynamics may differ and warrant examination in future research.

Looking ahead, the multilevel structure of the PFS opens important avenues for further exploration. Longitudinal studies could examine how state-level flow patterns accumulate over time and influence trait-like dispositions [20]. Experimental manipulations could also test how specific task features—such as feedback, challenge level, or autonomy—influence the emergence of each flow component [8]. Given the sensitivity of the PFS to within-person variability, it may also prove useful in intervention studies aimed at enhancing flow in domains such as education, sports, or work [4]. To establish score comparability across populations and languages, future work should pair formal content-validity procedures (expert relevance ratings, cognitive interviewing) with score linking/equating to the source instrument, followed by multi-group measurement-invariance tests and DIF analyses (e.g., across gender and age; cf. [32]). Methodologically, decomposing within- and between-person associations (e.g., latent state–trait or multilevel structural models), allowing for non-linear/time-varying trends, and exploring person-specific dynamics could sharpen inferences about how flow unfolds and stabilizes. In parallel, integrating physiological and behavioral indicators alongside self-reports would enable multimethod validation and clarify the psychophysiological signature of flow.

Taken together, our results show that the Polish PFS captures moment-to-moment flow with solid multilevel reliability and a coherent hierarchical structure. The scale is sensitive to temporal dynamics across repeated trials and exhibits the expected convergent pattern with established state and trait measures, while also highlighting selective—as opposed to uniform—trait–state correspondence. These features recommend the Polish PFS for experimental and applied settings where flow must be tracked within persons over time. Future work should extend validation to naturalistic contexts and integrate physiological and behavioral indices to further establish predictive validity.

## Supporting information

**S1 Table. Original and Polish versions of the Psychological Flow Scale (PFS) items.** Table lists each item's original (English) wording and Polish translation.
(DOCX)

**S1 Fig. Monte Carlo power analysis results.** Graphical summary of simulation-based power for the key analyses reported in the manuscript.

(JPEG)

## Author contributions

**Conceptualization:** Marcin Wojtasiński, Przemysław Tużnik, Tomasz Jankowski, Silvia Leoni.

**Data curation:** Marcin Wojtasiński, Przemysław Tużnik, Tomasz Jankowski, Silvia Leoni, Mateusz Chwaszcz, Dorota Miszczyszyn, Maria Banasik.

**Formal analysis:** Marcin Wojtasiński, Przemysław Tużnik, Tomasz Jankowski, Silvia Leoni.

**Funding acquisition:** Marcin Wojtasiński.

**Investigation:** Marcin Wojtasiński, Przemysław Tużnik, Tomasz Jankowski, Silvia Leoni, Dorota Miszczyszyn, Maria Banasik.

**Methodology:** Marcin Wojtasiński, Przemysław Tużnik, Tomasz Jankowski, Silvia Leoni.

**Project administration:** Marcin Wojtasiński, Przemysław Tużnik, Tomasz Jankowski, Silvia Leoni, Mateusz Chwaszcz, Dorota Miszczyszyn, Maria Banasik, Paweł Augustynowicz.

**Resources:** Marcin Wojtasiński, Przemysław Tużnik, Tomasz Jankowski, Silvia Leoni, Mateusz Chwaszcz, Paweł Augustynowicz.

**Software:** Marcin Wojtasiński, Przemysław Tużnik, Tomasz Jankowski, Silvia Leoni, Mateusz Chwaszcz, Paweł Augustynowicz.

**Supervision:** Marcin Wojtasiński, Przemysław Tużnik, Tomasz Jankowski, Silvia Leoni, Paweł Augustynowicz.

**Validation:** Marcin Wojtasiński, Przemysław Tużnik, Tomasz Jankowski, Silvia Leoni.

**Visualization:** Marcin Wojtasiński, Przemysław Tużnik, Tomasz Jankowski, Silvia Leoni.

**Writing – original draft:** Marcin Wojtasiński, Przemysław Tużnik, Tomasz Jankowski, Silvia Leoni, Dorota Miszczyszyn.

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
