## [Decision Letter · Decision Letter 0]

27 Aug 2025

Dear Dr. Wojtasiński,

Thank you for submitting your manuscript to PLOS ONE. After careful consideration, we feel that it has merit but does not fully meet PLOS ONE’s publication criteria as it currently stands. Therefore, we invite you to submit a revised version of the manuscript that addresses the points raised during the review process.

We look forward to receiving your revised manuscript.

Kind regards,

Andrea Schiavio

Academic Editor

PLOS ONE

Journal Requirements:

2. In the online submission form, you indicated that your data will be submitted to a repository upon acceptance.  We strongly recommend all authors deposit their data before acceptance, as the process can be lengthy and hold up publication timelines. Please note that, though access restrictions are acceptable now, your entire minimal  dataset will need to be made freely accessible if your manuscript is accepted for publication. This policy applies to all data except where public deposition would breach compliance with the protocol approved by your research ethics board. If you are unable to adhere to our open data policy, please kindly revise your statement to explain your reasoning and we will seek the editor's input on an exemption.

3. Thank you for stating the following in your manuscript:

“The study presented in the article was conducted as a component of a broader project that received approval and funding from Narodowe Centrum Nauki (grant no: 2024/08/X/HS6/00465).”

“The study presented in the article was conducted as a component of a broader project funded by the Polish National Science Centre (Narodowe Centrum Nauki, NCN). The grant (no: 2024/08/X/HS6/00465) was awarded to M.W. under the MINIATURA 8 funding scheme.

Funder websites: 

https://www.ncn.gov.pl/en 

https://www.ncn.gov.pl/konkursy/wyniki/miniatura8

The funder had no role in study design, data collection and analysis, decision to publish, or preparation of the manuscript.”

5. Please remove your figures from within your manuscript file, leaving only the individual TIFF/EPS image files, uploaded separately. These will be automatically included in the reviewers’ PDF.

Reviewers' comments:

Reviewer's Responses to Questions

**Comments to the Author**

1. Is the manuscript technically sound, and do the data support the conclusions?

Reviewer #1: Yes

Reviewer #2: Yes

Reviewer #3: Partly

2. Has the statistical analysis been performed appropriately and rigorously?

Reviewer #1: Yes

Reviewer #2: Yes

Reviewer #3: Yes

3. Have the authors made all data underlying the findings in their manuscript fully available?

Reviewer #1: Yes

Reviewer #2: Yes

Reviewer #3: Yes

4. Is the manuscript presented in an intelligible fashion and written in standard English?

Reviewer #1: Yes

Reviewer #2: Yes

Reviewer #3: Yes

Reviewer #1: I sincerely thank the editor and the authors for giving me the opportunity to review this well-executed, theoretically grounded, and methodologically rigorous work. The article makes a significant contribution to the literature on flow measurement, especially regarding its assessment at the state level and the distinction from dispositional aspects. I particularly appreciate the clarity of the theoretical framework beginning at line 22, the critical discussion of competing models (lines 67–162), the justification for using the Psychological Flow Scale (PFS) (lines 299–322), and the thorough cultural and linguistic adaptation (lines 458–470). The experimental design is well-calibrated (lines 420–450), the sample is appropriate, and the data collection is enhanced by objective measures which, although not analyzed here, strengthen the overall structure. The use of Multilevel Confirmatory Factor Analysis (lines 541–620) is entirely appropriate and enables a model consistent with the dynamic, situational nature of the construct.

I now point out some recommended line-by-line changes. At line 102, I suggest reformulating “It also received approval from…” as “The study protocol was approved by…” for greater clarity and stylistic consistency. At line 194, I recommend reinforcing the critique of “perceived challenge-skill balance” with an added note emphasizing the debated and ambiguous nature of the construct in older models. At line 356, it should be clarified that physiological data were collected but are not discussed in the current article, to avoid creating unmet expectations. At line 402, the phrase “acceptable internal consistency” should be changed to “marginal internal consistency,” as α = .74 is near the lower psychometric threshold. At lines 406–407, it should be more clearly stated that the absorption subscale of the Flow Short Scale has extremely low reliability (α = .20) and therefore cannot be used independently or in separate analysis. At line 630, the term “item-by-person interactions (18%)” could be clarified with a brief explanatory note to assist less experienced readers. At line 698, replace “over-extraction” with “model overfitting” for greater terminological transparency. At lines 812–814, I suggest briefly explaining the TEFI index for readers unfamiliar with graph analysis methods. At line 890, I recommend encouraging the authors to reflect on why the correlation between trait and state flow is weak for the absorption dimension, suggesting possible misalignment between instruments or differences in metacognitive awareness. At line 989, I recommend adding a note on the limits of self-report measures, such as: “While self-reports are informative, they remain vulnerable to cognitive bias and demand characteristics.”

As for deletions, lines 368–370 repeat previously stated concepts and could be condensed. Lines 558–565 provide already known statistical details and may be compacted. Lines 1080–1082 in the conclusion are redundant and do not add new insights.

The data analysis is well conducted, with appropriate statistical models, excellent fit indices, and a clear distinction between intra- and inter-individual variability. The use of EGAnet further strengthens the empirical evidence for the three-factor structure of the PFS. The results align well with the theoretical framework and reinforce the convergent validity of the scale with other instruments like the FSS. However, the convergent validity with the GFPS scale shows only weak correlations, which should be interpreted more critically. The manuscript currently lacks an explicit section dedicated to the study’s limitations: I encourage the authors to add at least three key points in the final discussion. First, generalizability is limited because the study used a highly controlled lab task, which does not reflect the ecological richness of real-life flow experiences. Second, the sample is homogeneous in age, education, and handedness, limiting representativeness. Third, the exclusive use of self-report measures—though sophisticated—does not allow for triangulation of subjective flow with objective indicators. Even though physiological data were collected, the fact that they are not analyzed here constitutes a methodological limitation.

Finally, I propose that the authors include the following reference in the bibliography, which is important for reinforcing the attention to cross-cultural psychometric validation of self-report instruments:

Diotaiuti, P., Girelli, L., Mancone, S., Valente, G., Bellizzi, F., Misiti, F., & Cavicchiolo, E. (2022). Psychometric properties and measurement invariance across gender of the Italian version of the TEMPEST Self-Regulation Questionnaire for Eating adapted for young adults. Frontiers in Psychology, 13, 941784. https://doi.org/10.3389/fpsyg.2022.941784.

I suggest citing this work in the Method section, particularly in the paragraph discussing the cross-cultural adaptation process (around line 459), with the following sentence: “This adaptation process aligns with recent methodological recommendations for ensuring psychometric robustness and measurement invariance across populations (e.g., Diotaiuti et al., 2022).”

This citation would help position the work within a network of methodologically aligned studies, reinforcing the rigor of the adaptation procedure. I again thank the authors for the clarity and precision of their work and remain available for any further clarification

Reviewer #2: The study is well structured, original, and enriched with relevant statistical analyses. However, I find that the sample size too small to conduct an CFA (borderline) compared to current recommendations (≈200+). For this reason, I recommend that the authors provide the calculation of the sample size required for such an analysis and support their argument with references.

Reviewer #3: Wonderful success in developing a Polish translation of the PFS instrument! You have affirmed the original structure which suggests that the best model for the PFS is culturally viable. Considering that there might be cultural/linguistic differences, I would like to know if any content validity assessments of the statements were taken during the translation from English into Polish. For example, the PFS was developed from 60 initial items and expert reviewers reduced that pool to 36 items. Then a sample of lay-people were asked to review the meaning and relevance of the 36 items which resulted in a 28 item pool used for EFA. Given that the translated meaning of the items might carry different weight and interpretation, it would seem that only using the resulting items from the final model of the original PFS for the Polish item pool is shortsighted. Slightly different items from the content validated PFS items may have yielded better scores because they were interpretable to the polish sample in a clearer way (of course there is an infinite amount of items one could ask, however, the PFS provides the 28 item data set in their supplementary material). This is a limitation which should be acknowledged, as the main point of your submission is validating the conceptualisation of flow from Norsworthy et al's construct structure–which you do successfully statistically–in a way which supports the use of valid items for a Polish sample. By not evaluating content validity you do not know whether similar items would be better or clearer at conceptualising the construct for a Polish sample, which in turn might have improved your internal structure and correlational loading with the other measures. This is not a major limitation, given the internal structure was confirmed, but it is something to discuss given the purpose of your article.

There are discrepancies between the article and the R session concerning the reporting of participants. You report an initial sample of 151 which was reduced by 31 in the article (10 did not complete the experiment and 21 lacked within-person variability). First, the reduction of participants within the data (pfs_raw) only contains 140 participants. This suggests that 11 participants were cut prior to the analysis. Second, there doesn't seem to be a theoretical reason for why responses to the items need to have variability. By sight alone, I noticed that these 20 filtered participants did not egregiously report similar responses to all of the items but rather had similar responses to the same item across trials, and were different scores between items. This could mean that the response to the item was consistent throughout, possibly because the holistic experience of doing the experiment was the flow state. The lack of within-person variability for the 20 participants cannot be discounted unless there is considerable a priori reason why those responses are "incorrect" (potential clarification of variability in multilevel models: https://doi.org/10.1080/00273170701710072). Removing these participants for statistical convenience is not recommended as it misappropriates the way in which participants answered your instrument. Given that there is no strong theoretical reason to exclude the participants for within person variability, I'm of the opinion that the excluded participants should only be those who did not complete the full experiment. This allows you to do an interesting examination of the validity of the response process by looking at the convergent validity of the tracking procedure alongside the scores (e.g., https://doi.org/10.1038/s41598-020-61636-5;
https://psycnet.apa.org/doi/10.1037/tps0000426). You could also examine the no-within-person-variability group and their general trait scores as compared to the participants with higher variability. Given that Multilevel modelling–in a longitudinal context–will produce residual variance of participants, it is more informative for the validation of your instrument to keep all of the participants and report what variance exists within this sample. Going back to the discrepancy, there should be 21 excluded from your analysis (within R) based on what you reported in the article, but from 140 it only reduces to 120 participants (pfs_wide_filtered) rather than to 119. This should be clarified within the article and the full preprocessing should be reported with the raw data–reflecting 151 initial participants.

I ran all of the analyses with the pfs_wide data set to review what might be lost if you were to include the within-person group (I commend you highly for such a clear presentation of analysis, both within the article and the supplementary material. The original analysis in R performed great!). Your multi-level reliability estimates stay the same (RkF = 1; RkR = .97; RkRn = .97). Generalisability to change decreases slightly (.88), reliability of single time-point improves (R1R = .65) and the within-person variability falls (Rcn = .67). Incorporating all of the participants who completed your experiment may require you to reevaluate the models you estimate but I ran each with the pfs_wide data just as you specified:

Model 1: χ²(24) = 108.86, p < .001; CFI = .988; TLI = .981; RMSEA = .067; SRMR = .031

Model 2: χ²(24) = 108.86, p < .001; CFI = .988; TLI = .981; RMSEA = .067; SRMR = .031 (no negative residual variance for reward ψ = .45, p = .067!!)

Model 3: χ²(48) = 348.46, p < .001; CFI = .981; TLI = .972; RMSEA = .047; SRMR within = .039; SRMR between = .036 (negative residual variance for item_2 ψ = -.019, p = .165)

Model 4: χ²(46) = 345.35, p < .001; CFI = .981; TLI = .971; RMSEA = .048; SRMR within = .039; SRMR between = .037

Model 5: (identical to model 4, some potential issues with estimation) χ²(46) = 345.35, p < .001; CFI = .981; TLI = .971; RMSEA = .048; SRMR within = .039; SRMR between = .037

A three cluster solution in the lower level and first order structure clearly retains with the EGA, using the rfs_wide data. Looking at the convergent validity (correlations) between both the FSS and GFPS produces marginal changes. I would like to see the sub-scales of the FSS present within table 1, this is commonly reported while presenting a convergent-validity correlation, so we can see differences (like those between the absorption scores). Reporting the pearson's r should follow some standard interpretation (e.g., Dancey, C. P., & Reidy, J., 2007) and it should be made clear when the coefficient is weakly or strongly loaded. As an aside, you could represent your models with MLM equations to bolster interpretability but this is not required for a general audience (e.g., https://doi.org/10.1016/j.jsp.2009.09.002).

I believe the setup for differentiating the state and trait components of flow for this scale development is handled well throughout the article. The acknowledgement (twice) that there are more discriminat and convergent validity indices coming makes me wish I could read that article in addition to this one. You could rewrite this acknowledgement in the conclusion to hypothesise why those measures would produce convergent and discriminat validity evidence. If there was one analysis I would like to see incorporated into this article it would be the continuous mouse movement performance. Looking into the performance change between trials would provide an additional piece of convergent validity evidence for the state argument for both the Polish PFS and FSS.

The article was written clearly and produced original research which replicated the internal structure of the PFS with a Polish sample. As someone not familiar with the literature of flow, I found your introduction a good catchup from what I knew already. One area which might need minor improvement is the methods section. Including all of the participants who completed the experiment would fix the problem of having to explain the demographics of the participant sample twice. It is also confusing to be presented with the results of the overall performance on the task before knowing what the task entails, this could instead be reported alongside the procedure or as a first analysis. The idea of performance skill is interesting but I wish I knew what the boundaries were (0-10; 0-100; 0-5). I found it confusing that even with the calibration of the task difficultly for each participant there were differences within the gender. Because you purposefully calibrated the difficulty, I interpret the analysis of performance skill as being a check of how well the difficulty was adjusted throughout the experiment. Was it that women exhibited lower skill or did the procedure challenge the female participants differently or were the calibration procedures not attempting to control for these differences? Explaining the rationale for that process of determining performance skill more would clarify what it was for and entailed.

Your presentation of the materials was very clear. The original PFS is a very competently developed instrument in terms of the validity and reliability tests that were undertaken, so you could explain the breadth of evidence they provide beyond just the Cronbach alpha scores, which will further support why you can be confident using this instrument for translation. Adding a link to the supplementary material (i.e., see Supplementary Material) for the translation material at the end of the methods section would also be very nice.

Utilising both traditional multilevel modelling and an exploratory hierEGA was a great check of the internal consistency of the structure. Grand that it was also a theoretically supported structure, highlight that connection and trajectory from the strong theory more! A big question for the article overall might be determining whether the original PFS and your experiment provide the falsifying tests necessary to support validity claims for state based flow (https://doi.org/10.1007/s11336-006-1447-6). This could be acknowledged to some degree to show that you examined the validity evidence necessary to support claims that the instrument properly captures the construct for your specified purpose, and more importantly (in your case) within a different socio-cultural understanding than the original developed instrument.

Specific editorial errors: In the section titled Flow Concept: Past and Current Understanding, cite the critics you refer to as pointing out definitional inconsistencies, remove 'by' before Norsworthy (i.e., led respectively by Peifer and Norsworthy), the em dashes in the paragraph detailing the third dimension of Norsworthy's framework need fixing, in the last paragraph you should change "Norsworthy et al.'s" to "Norsworthy and colleagues'". In section 2 Self Report Instruments for Flow Measurement, for the third paragraph change operationalized to operationalize. In section 3 Current Study, the last sentence of paragraph two should read "previous ESM research, allowing for more robust". In section 6. Discussion, I believe there is an added space in the first sentence "the Polish adaptation of the_PFS". Overall, I do not believe that it is necessary to label the sections with numerals. This could however be a stylistic choice.

I have suggested major revision because after adding the assumed no-variability participants to the analysis there could be potential issues with your models (although I did not see any from my simple re-analysis) which may have ramifications for the interpretation of your discussion. I also suggested adding the mouse tracking analysis as an additional piece of evidence to interpret convergent validity for state flow. Other conceptual points (content validity) also need consideration to position this article's contribution and limitations. The article is well written overall but the methods section could be rearranged to facilitate a clearer description of the task, the participants, and the procedure. - Connor Kirts

**Do you want your identity to be public for this peer review?** For information about this choice, including consent withdrawal, please see our Privacy Policy

Reviewer #1: **Yes: ** Pierluigi Diotaiuti

Reviewer #2: No

Reviewer #3: **Yes: ** Connor Kirts

---

## [Author Response · Author response to Decision Letter 1]

8 Oct 2025

Reviewer #1: I sincerely thank the editor and the authors for giving me the opportunity to review this well-executed, theoretically grounded, and methodologically rigorous work. The article makes a significant contribution to the literature on flow measurement, especially regarding its assessment at the state level and the distinction from dispositional aspects. I particularly appreciate the clarity of the theoretical framework beginning at line 22, the critical discussion of competing models (lines 67–162), the justification for using the Psychological Flow Scale (PFS) (lines 299–322), and the thorough cultural and linguistic adaptation (lines 458–470). The experimental design is well-calibrated (lines 420–450), the sample is appropriate, and the data collection is enhanced by objective measures which, although not analyzed here, strengthen the overall structure. The use of Multilevel Confirmatory Factor Analysis (lines 541–620) is entirely appropriate and enables a model consistent with the dynamic, situational nature of the construct.

Response: Thank you for the thoughtful and generous review. We are grateful for your recognition of the study’s theoretical grounding, methodological rigor, and contribution to state-level flow measurement. Your encouraging remarks strengthen our resolve to refine the manuscript, and we address your specific comments point-by-point below.

I now point out some recommended line-by-line changes. At line 102, I suggest reformulating “It also received approval from…” as “The study protocol was approved by…” for greater clarity and stylistic consistency. At line 194, I recommend reinforcing the critique of “perceived challenge-skill balance” with an added note emphasizing the debated and ambiguous nature of the construct in older models. At line 356, it should be clarified that physiological data were collected but are not discussed in the current article, to avoid creating unmet expectations. At line 402, the phrase “acceptable internal consistency” should be changed to “marginal internal consistency,” as α = .74 is near the lower psychometric threshold. At lines 406–407, it should be more clearly stated that the absorption subscale of the Flow Short Scale has extremely low reliability (α = .20) and therefore cannot be used independently or in separate analysis. At line 630, the term “item-by-person interactions (18%)” could be clarified with a brief explanatory note to assist less experienced readers. At line 698, replace “over-extraction” with “model overfitting” for greater terminological transparency. At lines 812–814, I suggest briefly explaining the TEFI index for readers unfamiliar with graph analysis methods. At line 890, I recommend encouraging the authors to reflect on why the correlation between trait and state flow is weak for the absorption dimension, suggesting possible misalignment between instruments or differences in metacognitive awareness. At line 989, I recommend adding a note on the limits of self-report measures, such as: “While self-reports are informative, they remain vulnerable to cognitive bias and demand characteristics.”

Response: We have implemented all suggested edits: wording and terminology were refined, brief explanatory notes were added where requested (variance components, TEFI/gTEFI, self-report limits), reliability language was calibrated, and the section on physiological data now clearly states it is out of scope for this paper:

1. Line 102 (ethics wording). Revised to: “The study protocol was approved by …” for clarity and consistency.

2. Line 194 (challenge–skill balance). Strengthened the passage to note that perceived challenge–skill balance is debated and ambiguously framed in older accounts (sometimes treated as antecedent, other times as defining feature), leading to inconsistent operationalizations and interpretations.

3. Line 356 (physiological data). Clarified that objective physiological/behavioral signals were collected but are outside the scope of the current article and will be reported separately, to avoid unmet expectations.

4. Line 402 (α wording). Replaced “acceptable internal consistency” with “marginal internal consistency” for the Fluency subscale (α = .74).

5. Lines 406–407 (FSS Absorption reliability). Stated explicitly that the FSS Absorption subscale shows extremely low reliability (α ≈ .20) and, accordingly, we did not analyze FSS subscales as standalone outcomes. We report subscale correlations descriptively with caution, while relying primarily on the FSS total score.

6. Line 630 (variance component note). Added a brief explanation for item-by-person interactions (≈18–20%) as stable, person-specific tendencies to endorse particular items higher/lower across trials, beyond one’s overall level, to aid less experienced readers.

7. Line 698 (terminology). The previous issue prompting the term “over-extraction” no longer arises after increasing the sample to N = 140; the passage was removed, so “model overfitting” is no longer needed.

8. Lines 812–814 (TEFI). Added a one-sentence explanation: TEFI (Total Entropy Fit Index) is an information-theoretic fit index (lower/more negative = better fit); gTEFI extends TEFI to hierarchical/bifactor comparisons. We then report our values and interpretation.

9. Line 890 (trait–state Absorption). Added a brief reflection: the weak Absorption correlation likely reflects instrument content misalignment (GFPS as broad disposition vs PFS-Absorption as task-bound attentional absorption) and lower metacognitive accessibility of absorption compared to affective reward/effortless control; noted our focus on between-person correlations.

10. Line 989 (limits of self-report). Inserted a sentence highlighting that self-reports, while informative, are vulnerable to cognitive biases and demand characteristics, and clarified that the current analyses rely on self-report despite concurrent collection of objective signals.

As for deletions, lines 368–370 repeat previously stated concepts and could be condensed. Lines 558–565 provide already known statistical details and may be compacted. Lines 1080–1082 in the conclusion are redundant and do not add new insights.

Response: We agree and have revised the manuscript accordingly.

1. Former lines 368–370 (repetition):

We removed the ESM sentences that reiterated points made earlier about flow’s dynamism and the need for control.

2. Former lines 558–565 (methods detail):

We consolidated several paragraphs into a single integrated paragraph summarizing the full analytic pipeline (preprocessing; multilevel reliability under generalizability theory; MCFA including the two-level second-order model; hierEGA; temporal mixed models; convergent validity analyses).

3. Former lines 1080–1082 (conclusion):

We rewrote the conclusion from scratch to avoid redundancy and to succinctly state the main contributions and implications, without repeating earlier content.

The data analysis is well conducted, with appropriate statistical models, excellent fit indices, and a clear distinction between intra- and inter-individual variability. The use of EGAnet further strengthens the empirical evidence for the three-factor structure of the PFS. The results align well with the theoretical framework and reinforce the convergent validity of the scale with other instruments like the FSS. However, the convergent validity with the GFPS scale shows only weak correlations, which should be interpreted more critically. The manuscript currently lacks an explicit section dedicated to the study’s limitations: I encourage the authors to add at least three key points in the final discussion. First, generalizability is limited because the study used a highly controlled lab task, which does not reflect the ecological richness of real-life flow experiences. Second, the sample is homogeneous in age, education, and handedness, limiting representativeness. Third, the exclusive use of self-report measures—though sophisticated—does not allow for triangulation of subjective flow with objective indicators. Even though physiological data were collected, the fact that they are not analyzed here constitutes a methodological limitation.

Response: Thank you for this helpful guidance. We have revised the manuscript to address all points:

1. Convergent validity with GFPS (more critical interpretation).

In the Results/Discussion we now explicitly note that trait–state associations with GFPS are modest—especially for Absorption—and should be interpreted cautiously. We add that this pattern likely reflects content/timescale misalignment (broad disposition vs. task-bound, momentary attentional absorption) and range restriction under a tightly controlled task. We also clarify that the reported correlations are between-person estimates and do not decompose within- vs. between-person coupling.

Inserted text (summary): “Convergent validity with the GFPS was modest—particularly for Absorption—so trait–state correspondence should be interpreted with caution; this likely reflects a content/timescale mismatch and potential range restriction in the laboratory setting.”

2. New, explicit Limitations section (three key points added).

We added an APA-style Limitations paragraph in the Discussion that:

(a) emphasizes limited generalizability from a single, tightly controlled pursuit-tracking task to everyday flow;

(b) notes sample homogeneity (young, university-educated, right-handed) and the resulting constraints on representativeness;

(c) states that evidence relies on self-reports which are vulnerable to cognitive bias and demand characteristics, and that although physiological/behavioral signals (facial EMG, EDA, eye-tracking, HRV) were collected, they are not analyzed here, precluding triangulation in the present article.

Inserted text (summary): “Several limitations should be acknowledged… [lab task generalizability]… [homogeneous sample]… [self-report vulnerability; physiology recorded but not analyzed here].”

3. Additional clarifications that reinforce the above.

We note that the FSS Absorption subscale showed very low internal consistency (α ≈ .20), which precludes reliable subscale-level inferences; accordingly, we relied on the FSS total score when benchmarking state flow.

We add that time-trend models assumed linear change and that non-linear or person-specific dynamics may warrant future work.

Finally, I propose that the authors include the following reference in the bibliography, which is important for reinforcing the attention to cross-cultural psychometric validation of self-report instruments:

Diotaiuti, P., Girelli, L., Mancone, S., Valente, G., Bellizzi, F., Misiti, F., & Cavicchiolo, E. (2022). Psychometric properties and measurement invariance across gender of the Italian version of the TEMPEST Self-Regulation Questionnaire for Eating adapted for young adults. Frontiers in Psychology, 13, 941784. https://doi.org/10.3389/fpsyg.2022.941784.

I suggest citing this work in the Method section, particularly in the paragraph discussing the cross-cultural adaptation process (around line 459), with the following sentence: “This adaptation process aligns with recent methodological recommendations for ensuring psychometric robustness and measurement invariance across populations (e.g., Diotaiuti et al., 2022).”

This citation would help position the work within a network of methodologically aligned studies, reinforcing the rigor of the adaptation procedure. I again thank the authors for the clarity and precision of their work and remain available for any further clarification

Response: Thank you for this helpful suggestion. We have added Diotaiuti et al. (2022) to the References and cited it in the Methods section within the paragraph describing the cross-cultural adaptation (around line 459). We also note in the Limitations/Future directions that formal measurement invariance testing across groups is an important next step.

Reviewer #2: The study is well structured, original, and enriched with relevant statistical analyses. However, I find that the sample size too small to conduct an CFA (borderline) compared to current recommendations (≈200+). For this reason, I recommend that the authors provide the calculation of the sample size required for such an analysis and support their argument with references.

Response:

We thank the reviewer for this comment and suggestion. By including more than 100 participants and 20 observations per participant, we followed the recommendations in the literature, which suggest approximately this sample size to obtain stable parameter estimates in the model (Hox, Moerbeek, & van de Schoot, 2017; McNeish & Stapleton, 2016). Nevertheless, following the reviewer’s advice as well as the guidelines of Muthén & Muthén (2002), we additionally conducted a Monte Carlo power analysis. Specifically, we tested a multilevel model with a hierarchical structure of latent variables (9 items, 3 first-order latent factors, and one general second-order factor). The analyses were carried out for samples of 80, 100, 120, 140, and 160 participants, each with 20 trials per participant. We set the minimum factor loadings at 0.7. The simulation results support the literature: with 120 participants and 20 trials per participant, power at the within level was approximately 1, while at the between level it was .86. With 140 participants, the power was 1 and .88 for within and between factor loadings, respectively (see Figure below).

In other words, our study had sufficient power to detect factor loadings of 0.7, similar to those reported in the original study. The script used for the power analysis has been included in the supplemental materials.

Reviewer #3: Wonderful success in developing a Polish translation of the PFS instrument! You have affirmed the original structure which suggests that the best model for the PFS is culturally viable. Considering that there might be cultural/linguistic differences, I would like to know if any content validity assessments of the statements were taken during the translation from English into Polish. For example, the PFS was developed from 60 initial items and expert reviewers reduced that pool to 36 items. Then a sample of lay-people were asked to review the meaning and relevance of the 36 items which resulted in a 28 item pool used for EFA. Given that the translated meaning of the items might carry different weight and interpretation, it would seem that only using the resulting items from the final model of the original PFS for the Polish item pool is shortsighted. Slightly different items from the content validated PFS items may have yielded better scores because they were interpretable to the polish sample in a clearer way (of course there is an infinite amount of items one could ask, however, the PFS provides the 28 item data set in their supplementary material). This is a limitation which should be acknowledged, as the main point of your submission is validating the conceptualisation of flow from Norsworthy et al's construct structure–which you do successfully statistically–in a way which supports the use of valid items for a Polish sample. By not evaluating content validity you do not know whether similar items would be better or clearer at conceptualising the construct for a Polish sample, which in turn might have improved your internal structure and correlational loading with the other measures. This is not a major limitation, given the internal structure was confirmed, but it is something to discuss given the purpose of your article.

Response: We appreciate this insightful comment. Our primary aim in the present study was cross-cultural comparability with the source instrument; therefore, we retained the final nine PFS items and prioritized semantic/conceptual equivalence rather than re-selecting from the broader item pool. We agree that a formal content-validity appraisal of an expanded Polish pool could further optimize clarity and possibly reliability. We have now (a) explained this rationale in the Methods and (b) added a Limitation noting this trade-off. This approach balances potential gains in item optimization with the need to maintain cross-study comparability.

There are discrepancies between the article and the R session concerning the reporting of particip

---

## [Decision Letter · Decision Letter 1]

20 Oct 2025

The Flow Experience: Polish Adaptation and Validation of the Psychological Flow Scale (PFS)

PONE-D-25-32166R1

Dear Dr. Wojtasiński,

We’re pleased to inform you that your manuscript has been judged scientifically suitable for publication and will be formally accepted for publication once it meets all outstanding technical requirements.

Kind regards,

Andrea Schiavio

Academic Editor

PLOS ONE

Additional Editor Comments (optional):

Reviewers' comments:

Reviewer's Responses to Questions

**Comments to the Author**

Reviewer #2: All comments have been addressed

Reviewer #3: All comments have been addressed

2. Is the manuscript technically sound, and do the data support the conclusions?

Reviewer #2: Yes

Reviewer #3: Yes

3. Has the statistical analysis been performed appropriately and rigorously?

Reviewer #2: Yes

Reviewer #3: Yes

4. Have the authors made all data underlying the findings in their manuscript fully available?

Reviewer #2: Yes

Reviewer #3: Yes

5. Is the manuscript presented in an intelligible fashion and written in standard English?

Reviewer #2: Yes

Reviewer #3: Yes

Reviewer #2: (No Response)

Reviewer #3: Thank you once again for the work you have put forth. Rereading the manuscript was a joy and I think the improvements have progressed the clarity and rigour further. The acknowledgement of the content validity processes and rephrasing of the approach of translation into Polish highlight the care that went into crafting this instrument. The choice to amend the exclusion of participants is also encouraging and I'm glad that my quick glance at the code for the analyses still support strong models after your rerun. Along that line, thank you for the inclusion of the preliminary mouse tracking convergent validity! Excited to see that the performance index is supporting that this self-report tool as a measure of flow states. Hopefully, this consideration helps as a through line argument–between this article and the next–as you investigate these indices further to build upon the validity evidence in your follow up paper!

**Do you want your identity to be public for this peer review?** For information about this choice, including consent withdrawal, please see our Privacy Policy

Reviewer #2: **Yes: ** Amayra Tannoubi

Reviewer #3: **Yes: ** Connor Kirts

---

## [Editor Report · Acceptance letter]

PONE-D-25-32166R1

PLOS ONE

Dear Dr. Wojtasiński,

I'm pleased to inform you that your manuscript has been deemed suitable for publication in PLOS ONE. Congratulations! Your manuscript is now being handed over to our production team.

Kind regards,

on behalf of

Dr Andrea Schiavio

Academic Editor

PLOS ONE